# Bartonella- and Borrelia-Related Disease Presenting as a Neurological Condition Revealing the Need for Better Diagnostics

**DOI:** 10.3390/microorganisms12010209

**Published:** 2024-01-19

**Authors:** Marna E. Ericson, B. Robert Mozayeni, Laurie Radovsky, Lynne T. Bemis

**Affiliations:** 1T Lab Inc., Gaithersburg, MD 20878, USA; mee@tlabdx.com (M.E.E.); brm@tlabdx.com (B.R.M.); 2Laurie Radovsky, M.D. LLC., St. Paul, MN 55102, USA; laurie@drradovsky.com; 3Department of Biomedical Sciences, Medical School Duluth Campus, University of Minnesota, Duluth, MN 55812, USA

**Keywords:** *Bartonella henselae*, *Bartonella vinsonii*, *Borrelia burgdorferi*, Lyme disease, Bartonellosis

## Abstract

The diagnostic tests available to identify vector-borne pathogens have major limitations. Clinicians must consider an assortment of often diverse symptoms to decide what pathogen or pathogens to suspect and test for. Even then, there are limitations to the currently available indirect detection methods, such as serology, or direct detection methods such as molecular tests with or without culture enrichment. *Bartonella* spp., which are considered stealth pathogens, are particularly difficult to detect and diagnose. We present a case report of a patient who experienced a spider bite followed by myalgia, lymphadenopathy, and trouble sleeping. She did not test positive for *Bartonella* spp. through clinically available testing. Her symptoms progressed and she was told she needed a double hip replacement. Prior to the surgery, her blood was submitted for novel molecular testing, where *Bartonella* spp. was confirmed, and a spirochete was also detected. Additional testing using novel methods over a period of five years found *Bartonella henselae* and *Borrelia burgdorferi* in her blood. This patient’s case is an example of why new diagnostic methods for vector-borne pathogens are urgently needed and why new knowledge of the variable manifestations of Bartonellosis need to be provided to the medical community to inform and heighten their index of suspicion.

## 1. Introduction

*Bartonella* spp. are fastidious, Gram-negative, intracellular bacteria, which are more prevalent than previously believed, often causing chronic infections [1]. *Bartonella henselae* is one of the most commonly detected *Bartonella* spp. in humans and is the causative agent in cat scratch disease (CSD), bacillary angiomatosis and bacillary peliosis [2]. However, 45 different species of *Bartonella* have been identified so far [3]. Bartonella has been found in blood-culture-negative endocarditis patients, more than 60% of whom presented without a fever [3,4]. Patients infected with *Bartonella* spp. present with variable symptoms, which may include but are not limited to skin lesions, weight loss, neurological symptoms, cardiac arrythmias, endocarditis, muscle and joint pain, fatigue, sleep disturbances, bladder pain, foot pain, anxiety and more [5]. A study on blood samples from 296 patients who had received a previous diagnosis of Lyme disease, arthritis, chronic fatigue syndrome, or other conditions revealed that most patients had evidence of *Bartonella* spp. infection using either serology, blood culture or polymerase chain reaction (PCR) [6]. Interestingly, 3 patients out of 296 were positive when using all three testing modalities [6]. More recently, a study of blood from 500 asymptomatic blood donors discovered that 20% harbored *Bartonella* spp., though no diagnostic method identified all of the positive samples [2]. Given the difficulty of diagnosing Bartonellosis based on its clinical presentation and current testing techniques, new and accurate diagnostics are needed.

Common diagnostic tests used to detect *Bartonella* spp. infections include serology, blood culture, and direct detection molecular assays such as the amplification of Bartonella specific sequences using PCR. Each of these diagnostic modalities has substantial limitations, leading to a major hurdle in the timely diagnosis and treatment of Bartonellosis [2]. A comprehensive review of studies describing serologic assays for Bartonella found that the use of these assays led to an underdiagnosis of Bartonella infection [7]. A significant fraction of patients with persistent Bartonella infection will have no detectable antibodies [6]. Serology requires species-specific antibodies, and thus may result in false results due to the large number of *Bartonella* spp., their cross-reactivity with other organisms and the variability of assays between clinical labs [8,9]. While blood culture is considered the definitive diagnostic techniques to identify many bacterial pathogens, *Bartonella* spp. are fastidious and slow growing, and therefore defy timely identification by blood culture [10]. Other options for identifying a Bartonella infection include molecular assays such as fluorescent in situ hybridization (FISH), indirect immunofluorescence and PCR of nucleic acids extracted from tissues or blood. PCR is the preferred diagnostic modality, but there is currently no consensus on the bacterial target sequences for amplification [2]. Thus, there is an urgent need for the development of new and accurate diagnostic methods that provide direct evidence of the infection.

Better treatment options for this disease will only be possible once adequate diagnostic tools are available, and ideally these tools will help to monitor and determine the endpoint of treatment. In this report, we describe the case of a patient who had symptoms of Bartonella infection following a suspected spider bite, but for whom commercially available molecular tests and serology were consistently negative. It was only when their blood was submitted for novel advanced molecular assays and FISH testing that the patient was confirmed to harbor *Bartonella henselae* and *Bartonella vinsonii*. An incidental finding led to the discovery that the patient was co-infected with *Borrelia burgdorferi*. The lack of reliable tests for vector-borne pathogens significantly delayed treatment, as it does for most patients suffering with the diverse symptoms of Bartonellosis.

## 2. Materials and Methods

### 2.1. Ethics Statement

The subject gave informed consent for inclusion before they participated in this study. The study was conducted in accordance with the Declaration of Helsinki, and the protocol was approved by the Advarra Institutional Review Board (Pro00049820), Columbia, MD, USA.

### 2.2. DNA Extraction from Blood

Whole blood was collected in EDTA (purple cap) and stored at 4 °C until processing. A 500 µL aliquot of blood was mixed with 500 µL of water for ten minutes at room temperature. Then, 600 µL of cell lysis buffer, RLTplus (Qiagen, Hilden, Germany), was added and the protocol for the AllPrep DNA/RNA kit was followed as described for the kit (Qiagen). The final step was to extract the DNA from the column in two washes with 50 µL of molecular-grade water.

### 2.3. PCR Primers and Positive Controls

The DNA extracted from blood collected at 5 months (Table 2) was amplified using a Hemi-Nested PCR protocol for *Bartonella* spp., essentially as previously published [11]. The DNA extracted at later times (month 36 and 60) were amplified via Touch Down PCR using the following protocols. The primers for the BRT1 gene from *Bartonella henselae* were BRT1 forward (5′-CCT GGA AGC TCT AAC ATC GAA CAC AGA ATA) and BRT1 reverse (5′-GGT CTG GAA GCA CTG ACA TCG AAT C). The primers for the plasmid gene, OspA from Borrelia burgdorferi, were OspA forward (5′-AAA CAG CGT TTC AGT AGA TTT GCC TGG TGA) and OspA reverse (5′-TTC AAG TGT GGT TTG ACC TAG ATC GTC AGA). The primers for the chromosomal gene, P13 from *Borrelia burgdorferi*, were P13 forward (5′-ATG AAA CTA GCA AGC AAG ATC CTA TTG TAC C) and P13 reverse (5′-GCC CTA TAC CAA CCG CAT CAA ATC). The primers for 18S ribosomal RNA from *Babesia* spp. were 18S *Babesia* spp. forward (5′-GAA GAC GAT CAG ATA CCG TCG TAG TCC TAA) and reverse 18S *Babesia* spp. (5′-TGG TGC CCT TCC GTC AAT TCC TTT AAG). All primers were sequenced and confirmed by means of the amplification of a positive control DNA sample from the pathogen and submission to ACGT Inc., Wheeling, IL, USA, for direct sequencing. The positive controls for *Bartonella henselae* and *Babesia microti* were purchased from the American Type Culture Collection, Manassas, Virginia (ATCC), namely *Bartonella henselae* DNA (49882D-5, ATCC) and *Babesia microti* DNA (PRS-398DQ, ATCC). The positive control for *Borrelia burgdorferi* was strain B31A3 (a kind gift from Dr. Jennifer Coburn, Medical College of Wisconsin, Milwaukee, WI, USA). One milliliter of bacteria growing in the stationary phase was collected by means of centrifugation and DNA was extracted using the Qiagen AllPrep DNA/RNA kit as described above.

### 2.4. PCR Methods and Direct Sequencing

Each PCR amplification started with Touch Down PCR (with half degree step down from 65 to 57 °C) followed by 40 cycles at 60 °C and a final extension at 72 °C for five minutes. GoGreen Taq Polymerase (Promega, Madison, WI, USA) was used for all PCR amplifications. The amplified PCR products were first confirmed via gel electrophoresis on 2% agarose gels. The water controls were determined to be free of contamination or primer dimers and the samples were shown to have a single amplicon of the expected size for each primer pair. The remainder of the sample was then purified using PCR Clean Up columns from Qiagen and cloned into T-vector cloning vector pCR2.1 TOPO provided in the TOPO cloning Kit (product number K4560-01, Life Technologies, Carlsbad, CA, USA). Plasmids were subjected to direct sequencing following submission to ACGT Inc., Wheeling, IL, USA for Sanger sequencing. Sequences were compared to the reported sequences for *Bartonella* spp., *Borrelia* spp. or *Babesia* spp. using the Blast program at the National Center for Biotechnology Information at the National Library of Medicine, USA. Sequences and accession numbers from Genbank are included in the Appendix A.

### 2.5. Imaging of Blood Smears and Buffy Coat Smears Following rRNA Fluorescent In Situ Hybridization (FISH)

Venous blood was collected in EDTA (purple cap), from which a small drop of whole blood was used to make a blood smear on a microscope slide. The remainder was processed via centrifugation and a small drop of the buffy coat was used to make another smear on a microscope slide. The blood and buffy coat smears were fixed with methanol and subjected to rRNA FISH. Blood and buffy coat smears were stained for *Bartonella henselae* and the nucleic acid dye DAPI. Images were 3.8 micron-thick Z-stack projections captured via single-photon microscopy on a Nikon A1R microscope (Nikon PLAN APO 60X/1.42 oil).

## 3. Case Presentation

In 2018, a 61-year-old female was hiking in Minnesota and experienced a painful bite, which was diagnosed as a spider bite based on the two large bite marks and the painful sensation when the bite occurred (Figure 1).

A large blue ring developed around the bite and the patient began to experience myalgia. One month after the bite, she was seen at her primary care clinic and was prescribed diclofenac sodium (Voltaren) 1% gel for arthritis pain. When her symptoms continued, she was prescribed doxycycline for two weeks, and the suspected arthritic pain decreased but did not resolve. A request for further antibiotic treatment was refused. Five months into her illness, her symptoms included blurry vision, lack of balance, muscle pain, night sweats and insomnia. Routine testing for the erythrocyte sedimentation rate, vitamin D levels, thyroid stimulating hormone and liver function were all within normal limits, as were tests for rheumatoid arthritis (cyclic citrullinated peptide and CCP3.1 IgA/IgG ELISA) and systemic lupus erythematosus (ANA). *Borrelia burgdorferi* IgG and IgM, *Bartonella henselae* and *Bartonella quintana* IgG and IgM, as well as parvovirus PCR, were all negative (Table 1).

Based on these negative test results, she was referred to the infectious diseases department for her ongoing symptoms and concerns regarding Lyme disease. The infectious disease physician stated that they “did not believe in persistent Lyme disease” and referred her to neurology for an MRI that came back negative except for a few small moderate nonspecific supratentorial white matter lesions.

At this time, her blood was used in a research study aimed at developing new PCR diagnostic techniques for *Bartonella* infections. The new diagnostic technique was based on a nested PCR method for the intergenic region of the 16S rRNA gene from Bartonella [11]. The PCR products were submitted for direct sequencing, which confirmed the presence of both *Bartonella vinsonii* and *Bartonella henselae* (Table 2 and Appendix A).

The patient took her PCR-positive research tests to a new physician known to treat chronic vector-borne illness. However, due to the developmental nature of the tests, the physician did not feel able to prescribe antibiotics. One year after the original spider bite, the patient was finally able to see another integrative medicine physician with a special interest in vector-borne diseases. Upon questioning, the patient reported no known exposure to ticks, lice, fleas or cats leading up to the spider bite. However, based on symptomology, commercially available testing was ordered for a comprehensive list of pathogens. At the time of this visit and blood draw, all clinically available testing was negative (Table 1).

The community of progressive physicians who work to help such patients are familiar with the lack of accurate diagnostic techniques for vector-borne disease and often empirically treat based on clinical presentation. Based on her symptoms consistent with Bartonella infection, the patient was prescribed clarithromycin and rifampin treatment based on the known antibiotic sensitivity of *Bartonella* [12]. Despite these antibiotics, she continued to decline. Due to ongoing neurological symptoms, her primary care clinic referred her for evaluation at clinics specializing in multiple sclerosis (MS) and amyotrophic lateral sclerosis (ALS), both of which were ruled out. She was then offered psychological counseling and physical therapy. Her physical therapist recommended a wheeled walker in the summer of 2020. In October of 2020, she was told that she should think about taking a vacation with family as she probably would not be able to do so in the future. She reported to multiple physicians that her hips sounded like popcorn whenever she walked or climbed stairs. In early 2021, her physical therapist recommended she use a wheelchair. At this point, her primary care doctor ordered hip X-rays (Figure 2).

The X-rays revealed bilateral hip degeneration which required surgical replacement of the joints. Prior to surgery, she submitted a blood sample to TLab Inc. for *Bartonella henselae* rRNA FISH molecular testing with enhanced sensitivity using a confocal microscope system (Figure 3).

The findings at that time again confirmed that she had a *Bartonella* infection and incidentally, a spirochete was found in a buffy coat smear (Figure 4).

The positive results of the FISH studies with a probe specific for *Bartonella henselae* (23S rRNA) confirmed that the patient was positive for this organism. However, of equal interest was the detection of a spirochete-like organism (Figure 4). Given her symptomology and the known possibility of co-infections in Lyme disease, the spirochete was suspicious for *Borrelia burgdorferi*. To confirm this hypothesis, DNA was extracted from whole blood and was confirmed to be positive for *Borrelia burgdorferi* via PCR and direct sequencing (Figure 5, Table 1, 36 months post bite). Once treatment for Borrelia was initiated, the patient began to improve.

Since then, the patient has been on antibiotics intermittently; when she ceases taking them, her symptoms recur within 3 months. Her blood was again tested 5 years after the spider bite and remains positive for both *Borrelia burgdorferi* and *Bartonella* spp. (Appendix A).

## 4. Discussion

This case report illustrates the inadequacy of conventional tests in diagnosing *Bartonella* spp. infections, and the potential promise of enhanced techniques. Part of the difficulty in diagnosis stems from the plethora of symptoms that patients may present with, making it difficult for clinicians to consider Bartonellosis in the differential diagnosis [5,6,13,14,15,16]. An additional area of confusion is the wide variety of vectors able to transmit *Bartonella* spp., including but not limited to fleas, ticks and sand flies. While the reservoir host for some *Bartonella* spp. is humans, many other mammalian species can also have chronic bacteremia and transmit their infection to humans via insects that feed on both species [1,6]. It is likely that a One Health approach (the recognition that human health, animal health, and ecosystem health are inextricably linked) to Bartonella infections should be considered, as family pets may be incidental hosts [17].

In the case presented here, the patient had a painful bite which was diagnosed as a spider bite. While there is circumstantial evidence that *Bartonella* spp. are transmitted by arthropod vectors including fleas, flies and ticks [5,18], this remains controversial [19]. A case report of *Borrelia burgdorferi* infection in a child with a spider bite dismissed the possibility that this bite was the cause, preferring instead to blame an unnoticed tick bite [20]. In another case report, a family was diagnosed with *Bartonella* spp. (based on positive serology) following an invasion of their home by woodlouse hunter spiders [21]. Woodlouse hunter spiders and their prey, the woodlouse, were captured near the family’s home, and both organisms were found to harbor *Bartonella henselae* and/or *Bartonella vinsonii* [21]. Despite these reports, there are no studies proving the transmission of *Bartonella* spp. or *Borrelia* spp. by spiders.

Regardless of the identity of the vector, patients experience a wide range of symptoms and may go undiagnosed for years. Undetected and untreated *Bartonella* and *Borrelia* co-infections place patients at risk of long-term illness, which has been documented for more than 20 years [22,23]. The patient described in this report experienced continuous pain in both hips and degradation that went undiagnosed because her symptoms were attributed to possible MS or ALS, even though *Bartonella* and *Borrelia* have previously been detected in the bones and joints from patients with extensive hip involvement requiring joint replacement [24,25]. At the time we began testing, the patient appeared asymptomatic for an infectious disease because she was afebrile. Similarly, lack of a fever has been noted to contribute to the delayed diagnosis of *Bartonella* spp. infection in a subset of endocarditis cases [5].

Novel methods are needed to detect vector-borne pathogens in blood and tissue samples. The advantage of using FISH on blood smears is that minimal processing of the specimen is needed; no sample manipulation is required that could compromise the test. Improving the PCR methods and FISH-based microscopic techniques described here may prove helpful in such patients. Extracting DNA from whole blood allowed for the PCR confirmation that she had *Bartonella henselae* and *Bartonella vinsonii*, while the novel RNA FISH assay has the added advantage of allowing the identification of other co-infections as well. This is possible because DAPI stains cellular DNA, including that of the spirochete (Figure 4). This confirmed that the patient had Lyme disease as well as *Bartonella* co-infection and allowed for a change in antibiotic regimen that resulted in an improved outcome, since co-infections in Lyme disease are known to complicate treatment [5,22].

The CDC currently recommends a two-step test for the diagnosis of Lyme disease, where both steps must be positive for antibodies to *Borrelia burgdorferi* [26]. Although the specificity of this testing method is high, the sensitivity is low and inadequate for a medical diagnostic test [27]. Approved testing guidelines for *Bartonella* spp. are somewhat ill defined, with the CDC suggesting the use of a variety of modalities including serology, PCR, blood culture or lymph node biopsy depending on the clinical presentation [28]. There is a pressing need for new US Food and Drug Administration-approved tests for both Borrelia and Bartonella. A point-of-care test would be ideal in the future [29].

Serology and other antibody-based tests are usually used for *Bartonella* and *Borrelia* detection in the United States because they are available and because healthcare professionals are often unaware of the limitations of serology—that it is an indirect assessment of an immune response that may vary greatly among patients and over time in individual patients regardless of treatment. In this patient, all commercially available testing was negative (Table 1). She never tested positive through serology for *Bartonella* spp., even though she was clearly FISH- and PCR-positive. The limitations of serology for detecting an active infection need to be more clearly understood by the medical community. For example, in the United Kingdom, serological testing for *Bartonella* is no longer used as a diagnostic technique for *Bartonella* spp. because serology is unable to distinguish an ongoing infection from a previous infection [30].

In contrast, multiple tests using either of the research-use RNA FISH or PCR were positive for *Bartonella henselae* in this patient. Once the buffy coat smear from TLab, Inc showed an incidental finding of a spirochete-like organism, we were able to confirm that it was most likely *Borrelia burgdorferi* because two PCR assays targeting two genes from *Borrelia burgdorferi*, OspA and P13, were positive (Table 2). Despite the intermittent use of antibiotics for five years, the patient remains positive for *Bartonella henselae* and *Borrelia burgdorferi* (Table 1, 60 months). Improved diagnostic tests have provided direct evidence that her infections persist.

This case report illustrates several important gaps in our knowledge of the persistence of *Bartonella* spp. and co-infecting pathogens like *Borrelia burgdorferi*. New diagnostic methods and new understanding by medical professionals are needed to prevent the long journey that patients experience to find care for co-infections. One of the most important needs is for more sensitive diagnostic tests for Bartonella and Borrelia infections. When new diagnostic techniques for these two pathogens become generally available to the medical community and to their patients, there will be an improved understanding of the related diseases and the possibility to use such tests as biomarkers in clinical trials to develop effective treatments for these devastating persistent infections.

## Figures and Tables

**Figure 1 microorganisms-12-00209-f001:**
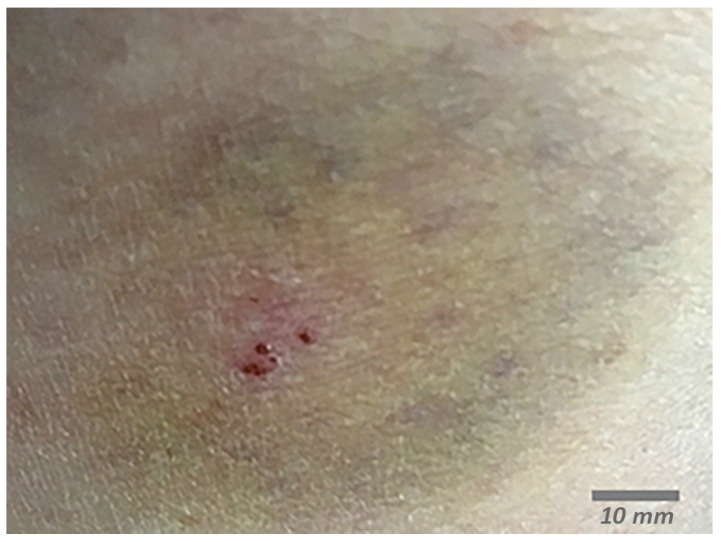
Self-captured cell phone image of bite site, 4 days after the bite. Blue ring appeared at 24 h post bite and turned into a bruise at Day 4.

**Figure 2 microorganisms-12-00209-f002:**
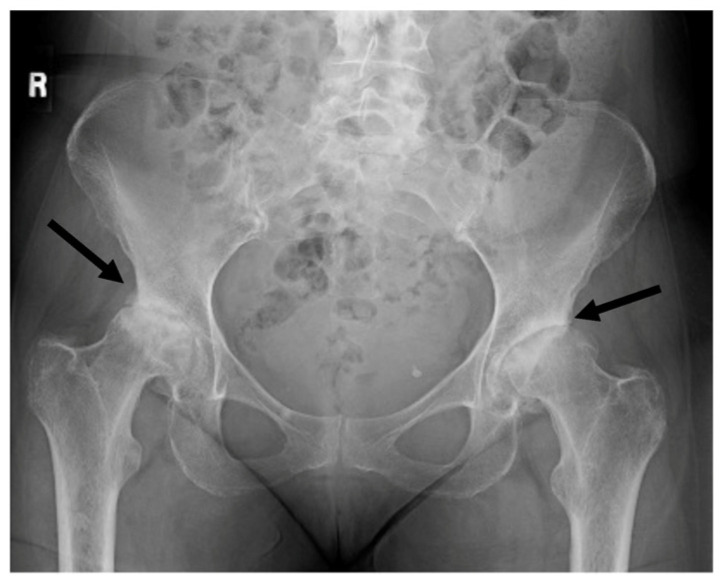
X-rays taken in the spring of 2021, two and a half years after the spider bite. No symptoms of arthritis had been observed prior to this event and the patient had been followed closely for signs of arthritis due to her family history and her participation in a long-term research study for the risk of rheumatoid arthritis. Arrows indicate joint space narrowing indicating loss of cartilage.

**Figure 3 microorganisms-12-00209-f003:**
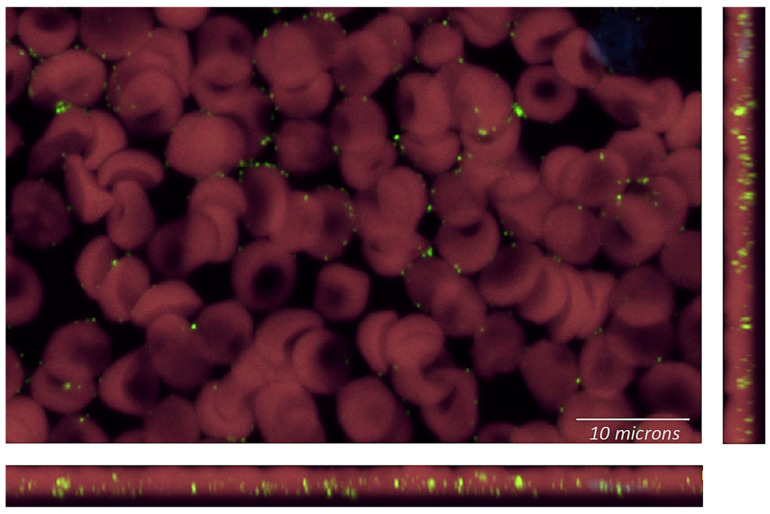
Blood smear stained with a fluorescently tagged RNA probe (FISH) for *Bartonella henselae.* Endogenous fluorescence, captured in the green and red channels, of bi-concave red blood cells, which are pseudo-colored red. The *B. henselae* RNA FISH signal is pseudo-colored green. This image is a 3.8 micron-thick Z-stack projection captured via single-photon confocal microscopy (Nikon PLAN APO 60X/1.42 oil).

**Figure 4 microorganisms-12-00209-f004:**
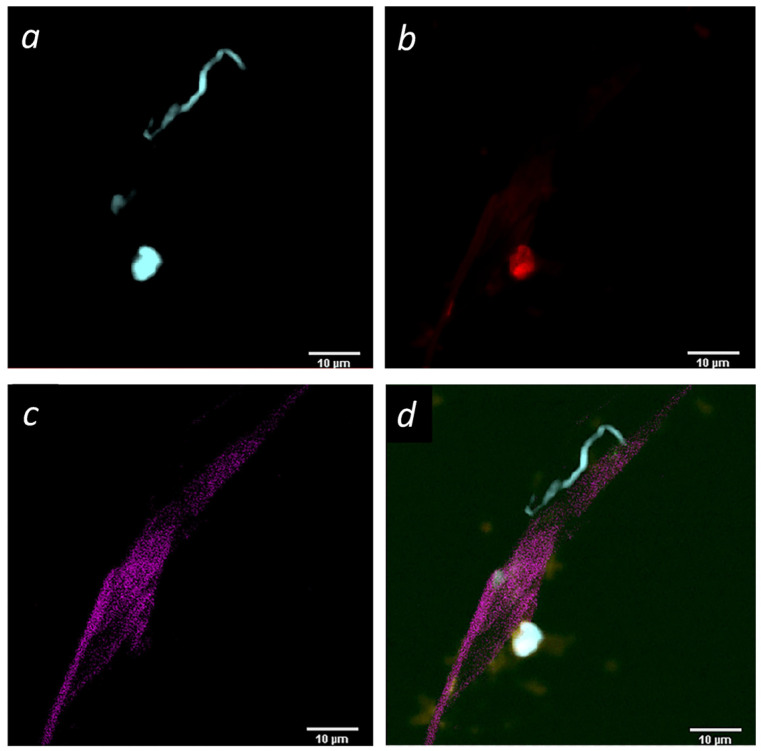
*Bartonella henselae* and a spirochete-like organism were both detected in this buffy coat smear sample. The smear of the isolated buffy coat reveals (**a**) the DAPI stain of nucleotides (pseudo-colored aqua); (**b**) the fluorescent-tagged RNA probe (FISH) for *Bartonella henselae* (pseudo-colored red); (**c**) the endogenous fluorescence of the possible biofilm structure captured in the far-red channel (pseudo-colored magenta) and (**d**) the merged image. Image is a 3.8 micron-thick Z-stack projection captured via single-photon confocal microscopy (Nikon A1R, 60X PLAN APO 60X/1.42 oil objective).

**Figure 5 microorganisms-12-00209-f005:**
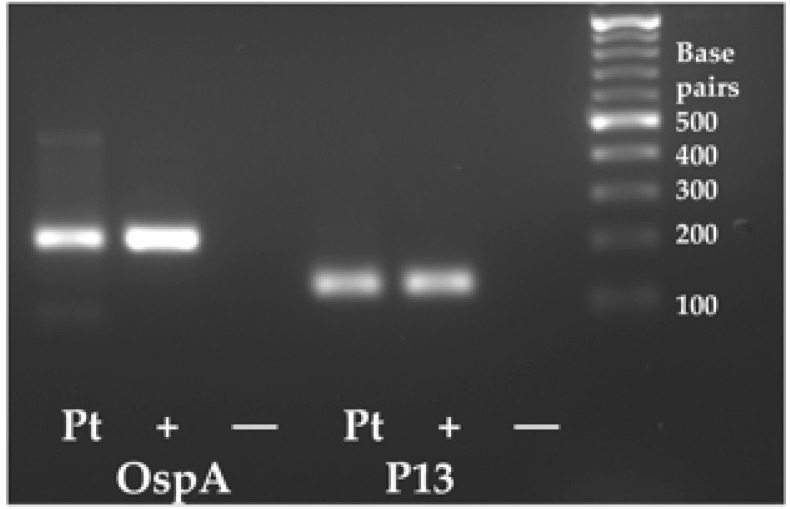
PCR amplification of DNA extracted from whole blood was positive for *Borrelia burgdorferi*. PCR primers were designed for small amplicons of genes present in a plasmid of *Borrelia burgdorferi* (OspA, 220 bp) and for P13 (135 bp), a chromosomal gene from the genome of *Borrelia burgdorferi*. Pt is the patient sample, (+) is a positive control DNA extracted from *Borrelia burgdorferi*, and ‘—’ in the third column is a PCR water control with no DNA added. Amplicons were sequence confirmed (Appendix A).

**Table 1 microorganisms-12-00209-t001:** Diagnostic testing for vector-borne pathogens.

Clinical Diagnostic Test	Months after Bite	
	5 Months *	12 Months *
*Bartonella henselae* IgG Ab	Negative	
*Bartonella henselae* IgM Ab	Negative	
*Bartonella quintana* IgG Ab	Negative	
*Bartonella quintana* IgM Ab	Negative	
*Parvovirus* B19 PCR	Negative	
Lyme disease was not suspected due to the absence of a tick bite	Not tested	
*Bartonella henselae* IgG/IgM by (ELISA)		Negative
*Bartonella henselae* PCR		Negative
*Bartonella clarridgeiae* PCR		Negative
*Bartonella bacilloformis* PCR		Negative
*Bartonella elizabethae* PCR		Negative
*Bartonella quintana* PCR		Negative
Lyme disease Western blot (*Borrelia burgdorferi*) IgG/IgM By CDC or alternative criteria		Negative
*Borrelia miyamotoi* PCR		Negative
*Borrelia turicatae* PCR		Negative
*Borrelia mayonii* PCR		Negative
*Borrelia hermsii* PCR		Negative
*Borrelia parkeri* PCR		Negative
*Babesia microti* IgG/IgM by IFA		Negative
*Babesia microti* IgG/IgM		Negative
*Babesia microti* PCR		Negative
*Babesia duncani* PCR		Negative

* All blood tests at 5 months were submitted to the clinical labs at the Mayo Clinic, Rochester, MN, USA. All blood tests at 12 months were submitted to Medical Diagnostics Laboratories, LLC, Hamilton, NJ, USA.

**Table 2 microorganisms-12-00209-t002:** Novel molecular assays for *Bartonella* spp. and *Borrelia burgdorferi*.

Pathogen and Novel Diagnostic Test	Months after Bite		
	5 Months	36 Months	60 Months
*Bartonella henselae* PCR	Positive (16S intergenic)	Positive (BRT1)	Positive (BRT1)
*Bartonella vinsonii* PCR	Positive (16S intergenic)		
*Borrelia burgdorferi* PCR		Positive (OspA)	Positive (OspA)
*Babesia microti* PCR			Negative (18S)

## Data Availability

Data are contained within the article or Appendix A.

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
