# Peer review of "Bartonella- and Borrelia-Related Disease Presenting as a Neurological Condition Revealing the Need for Better Diagnostics"

_microorganisms, 2024, doi:10.3390/microorganisms12010209_

Round 1

Reviewer 1 Report

Comments and Suggestions for Authors

The article presents an interesting point of view on the difficulty of diagnosing the diseases mentioned and I suggest some improvements and clarifications if it is to be accepted by the journal.

line 40-41: If diagnosis is so difficult, why, interestingly, are "only" three patients positive in all modalities? That's confusing, I suggest removing the "only". 

line 44: it is.

line 56: Is this case report from UK? This information is lost in the middle of the text. Also, it's the first time you are pointing UK, so you must write the whole name first, then abbreviations. 

line 60: "...PCR of nucleic acids extracted from tissue or blood. PCR is the preferred" it's repetitive, please rephrase. 

line 70: At this point of your text, you did not say it was a woman patience, so you must be impartial talking about the blood sample.

lines 91-93: Which tests? You must describe in the text too.

Table 1: - IgG and IgM is what are you looking for from a serology test, so you need to include the type of test, or explain in the text. - Please include which genes you target, or which primers were used for this PCR reactions, or at least respective references

Lines 103-104: Which PCR products? From the study cited. Specify. Also, you need to provide data about the sequencing. How it was it done? Which program/site the sequences where compared? 

Sup. Data: You must include references. Where are those access numbers from?

Line 132: Which molecular test? You must include in text.

Line 143-144: Why, after seeing a spirochete shape, hasn't the same blood been tested for darkfield smears, where it might be possible to see mobile spirochete shapes and perhaps confirm this hypothesis?

Line 158: Again, you must include in the text: which genes, and respective references.

Fig 5: Why was it used just these 2 primers? You used the same species you were looking for as a positive control, and in the gel, they are still next to each other, how can you confirm that this is not contamination?

Line 177: Point which are those vectors.

Line 179: How can they pass the infection to a human?

Line 185: I suggest remove the phrase "the transmission by arachnids is even less accepted", and finalize this paragraph saying there is NO evidence of Bartonella transmission by arachnids in US, and also NO evidence of Borrelia transmission by spiders... "Despite these reports, there are no studies proving the transmission of Bartonella by arachnids, nor of Borrelia by spiders."

Lines 195-197: I suggest removing this paragraph and finalizing the previous one as suggested above. 

Line 208: You did not describe how the method works; you just used the method. Should be interesting explain about it. 

Line 217-220: Rephrase, it's confusing. 

Line 223: Again, why UK? If you are just pointing a data, your phrase should not start with this, you could change for "As serology is unable to .... , it has been reconsidered as a diagnostic method, for example in the UK, where serological test is no longer available."

Line 228: "showed an incidental finding of Borrelia burgdorferi." This is not correct; it was an incidental finding of a "spirochete-like". You cannot confirm this is a spirochete just by that image.

Line 231: "Improved diagnostic tests have provided direct evidence 231 that her infections persist." Based in our results I suggest remove this sentence, also you already said that in the phrase before: " the patient remains positive for Bartonella spp. and Borrelia burgdorferi..."

As a suggestion: It would be interesting to point out whether during the assessments and based on the results presented, this patient reported contact with ticks, body louse, or cats. Or if this was not questioned, it could be discussed in the clinical approach section.

Reviewer 2 Report

Comments and Suggestions for Authors

The authors presented a very important case of Bartonella and Borrelia infection, in which they show diagnostic problems. The most important diagnostic data are presented in Table 1, which shows that the level of antibodies in infected people may be negative and only molecular tests confirm the diagnosis. The authors also pay attention to the development of research methods and their role in improving diagnostics. The results are confirmed by 5 photos.

I propose to publish the article after adding a description of the current international guidelines (with references) regarding the diagnosis of Bartonella and Borrelia in the Discussion.

Author Response

Thank you for your helpful comments we have added the following paragraph and several references and websites that we used in the revision.

The CDC currently recommends a two-step test for the diagnosis of Lyme disease, where both steps must be positive for antibodies to Borrelia burgdorferi [26]. Although, the specificity of this testing method is high, the sensitivity is low and inadequate for a medical diagnostic test [27]. Approved testing guidelines for Bartonella spp. are somewhat ill defined, with the CDC suggesting the use of a variety of modalities including serology, PCR, blood culture or lymph node biopsy depending on the clinical presentation [28]. There is a pressing need for new US Food and Drug Administration approved tests for both Borrelia and Bartonella. A point of care test would be ideal in the future [29].

 CDC website. Available online at: https://www.cdc.gov/lyme/diagnosistesting/index.html (accessed on 14, January 2024)

Stricker RB, Johnson L. Lyme disease diagnosis and treatment: lessons from the AIDS epidemic. Minerva Med 2010, 101(6), 419-25. PMID: 21196901.

CDC website. Available online at: https://wwwnc.cdc.gov/travel/yellowbook/2024/infections-diseases/bartonella-infections#diagnosis (accessed on 14, January 2024)

Marques AR. Laboratory diagnosis of Lyme disease: advances and challenges. Infect Dis Clin North Am. 2015, 29(2), 295-307. doi: 10.1016/j.idc.2015.02.005. PMID: 25999225; PMCID: PMC4441761.

Round 2

Reviewer 1 Report

Comments and Suggestions for Authors

The manuscript presents significant changes since the first version, interesting data and organized presentation, and seems to be ready for publication.